# In Vitro Entero-Capillary Barrier Exhibits Altered Inflammatory and Exosomal Communication Pattern after Exposure to Silica Nanoparticles

**DOI:** 10.3390/ijms20133301

**Published:** 2019-07-05

**Authors:** Jennifer Y. Kasper, M. Iris Hermanns, Annette Kraegeloh, W. Roth, C. James Kirkpatrick, Ronald E. Unger

**Affiliations:** 1Institute of Pathology, University Medical Center, 55131 Mainz, Germany; 2INM, Leibniz Institute for New Materials, D-66123 Saarbrücken, Germany

**Keywords:** intestinal microvasculature, inflammatory bowel disease, intestinal barrier in vitro, Caco-2, ISO-HAS-1, soluble E-selectin, sICAM-1, exosomes, silica nanoparticles, SIRS

## Abstract

The intestinal microvasculature (iMV) plays multiple pathogenic roles during chronic inflammatory bowel disease (IBD). The iMV acts as a second line of defense and is, among other factors, crucial for the innate immunity in the gut. It is also the therapeutic location in IBD targeting aggravated leukocyte adhesion processes involving ICAM-1 and E-selectin. Specific targeting is stressed via nanoparticulate drug vehicles. Evaluating the iMV in enterocyte barrier models in vitro could shed light on inflammation and barrier-integrity processes during IBD. Therefore, we generated a barrier model by combining the enterocyte cell line Caco-2 with the microvascular endothelial cell line ISO-HAS-1 on opposite sides of a transwell filter-membrane under culture conditions which mimicked the physiological and inflamed conditions of IBD. The IBD model achieved a significant barrier-disruption, demonstrated via transepithelial-electrical resistance (TER), permeability-coefficient (P_app_) and increase of sICAM sE-selectin and IL-8. In addition, the impact of a prospective model drug-vehicle (silica nanoparticles, aSNP) on ongoing inflammation was examined. A decrease of sICAM/sE-selectin was observed after aSNP-exposure to the inflamed endothelium. These findings correlated with a decreased secretion of ICAM/E-selectin bearing exosomes/microvesicles, as evaluated via ELISA. Our findings indicate that aSNP treatment of the inflamed endothelium during IBD may hamper exosomal/microvesicular systemic communication.

## 1. Introduction

Multicellular in vitro barrier models are increasingly being used in studies to evaluate, e.g., immune responses of the intestinal barrier after exposure to externally added stimuli. In vitro models with enterocytes combined with several other tissue-relevant cell-types such as monocytes/macrophages or dendritic cells [1,2,3,4], intraepithelial lymphocytes [5], fibroblasts [6] or with enteric glial cells [7,8] have been described. However, until recently, the intestinal microvasculature (iMV) has been largely ignored. As a semipermeable diffusion barrier and a second line of defense for pathogens and compounds, the iMV is a crucial participant in the innate immunity observed in the gut. The incorporation of endothelial cells into an in vitro enterocyte barrier model would give insight into the cell communicative processes and the gut barrier integrity, especially under inflammatory conditions that are observed in inflammatory bowel disease (IBD) [9].

The intestinal microcirculation with its endothelial lining is also being considered as a therapeutic point in IBD, targeting, among others, leukocyte adhesion processes, ICAM-1 (cellular adhesion molecule 1) and E-selectin, whose expression patterns are induced in IBD [10]. Other studies have shown that the serum levels of the soluble forms of these adhesion molecules, sICAM-1 and sE-selectin, which are cleaved from the cell membrane, are elevated in IBD patients during inflammation. Therefore, these soluble forms are considered as serum biomarkers defining the severity of active IBD [11]. On the other hand, circulating ICAM and E-selectin could also represent ICAM/E-selectin bearing exosomes or microvesicles. Activated endothelial cells release microvesicles (ca. 100 nm to 1 µm in diameter) that bear ICAM-1 and E-selectin on their surface. These endothelial microparticles (EMP) may serve as potential biomarkers for vascular injury. They can exhibit pro- as well as anti-inflammatory functions [12].

A number of in vivo models to study iMV behavior upon systemic inflammation have been described using Wistar rats [13]. In vitro studies examining ICAM/E-Selectin induction in monocultures of primary isolated intestinal microvascular endothelial cells (IMEC) have also been described [14]. However, to date, it appears that only a single in vitro coculture model mimicking the intestinal barrier consisting of microvascular endothelial cells in combination with enterocytes has been described [15]. No in vitro model for IBD has been developed. Thus, a model mimicking the in vivo intestinal entero-capillary barrier would be highly useful for understanding the exacerbated inflammatory progressions that result in barrier disruption in IBD patients. Furthermore, in vitro models mimicking disease situations can be indispensable in screening and evaluating the efficacy of novel therapeutic strategies such as nanoparticle-mediated drug delivery which specifically targets the endothelial lining [16].

In this study, we have used and optimized the intestinal blood barrier model Caco-2/ISO-HAS-1 that has been previously described [15]. The enterocyte cell line Caco-2 and the microvascular endothelial cell line ISO-HAS-1 were seeded on opposite sides of the same semipermeable transwell filter membrane. We refined this model and achieved a tight and stable barrier mimicking a “healthy” state that was monitored via transepithelial electrical measurements (TER) and by the transport assay with NaFlu (Sodium-Fluorescein). Furthermore, a distinct stimulation procedure using inflammatory mediators was developed to simulate a barrier disruption and inflammation as is observed in vivo during IBD. The production of inflammatory mediators (IL-8, soluble ICAM-1 and soluble E-selectin) was determined via ELISA. In addition, we applied different surface-modified amorphous silica nanoparticles (aSNP) as putative drug vehicle models to the endothelium during ongoing inflammation to examine their impact on the inflammatory responses. Finally, exosomal release was evaluated after exposure to inflammatory mediators and/or aSNPs and correlated to soluble ICAM-1 and soluble E-selectin release from cells. Surprisingly, after total exosomal isolation of the culture supernatants, we found that the endothelial lining released ICAM/E-selectin positive EMPs, and that this release was significantly reduced after aSNP treatment.

## 2. Results

Figure 1 depicts the immunofluorescent staining of β-Catenin of cell types Caco-2 and ISO-HAS-1 in coculture. In the untreated controls, well-defined cell contacts (adherens junctions) were observed. The stimulation with LPS on the apical side did not cause a morphological change to either of the cells based on the visual staining pattern of β-Catenin. However, after the stimulation with LPS on the apical side and 3C (cytokine mixture: IL-1β (100 U/mL), TNF-α (600 U/mL) and IFN-γ (200 U/mL)) on the basolateral side, the β-Catenin signal appeared inconsistent and more diffuse in ISO-HAS-1, and partly decreased in Caco-2. The application of LPS to the basolateral endothelial side also showed a diffuse staining pattern of β-Catenin for the ISO-HAS-1, but not for Caco-2. Exposure of 3C to the endothelial side (basolateral) resulted in a loss of β-Catenin staining in distinct areas of the Caco-2 monolayer, whereas ISO-HAS-1 appeared unchanged based on the β-Catenin staining. The treatment with LPS and 3C on the endothelial side resulted in a diffuse staining pattern on the endothelial side, but the Caco-2 cells appeared unaffected. In all cases, the monolayers of both cell types remained intact. Surprisingly, a reduced β-Catenin fluorescence signal was observed after treatment with the combined LPS on the apical and the 3C on the basolateral side. Under these conditions the TER values demonstrated a recovery of the barrier disruption caused by the addition of 3C alone.

Figure 2 illustrates the transepithelial electrical resistance TER (Figure 2A) and the apparent permeability coefficient (P_app_, Figure 2B) of the coculture CC Caco-2/ISO-HAS-1 after the addition of various compounds to the apical (ap; epithelial) and basolateral (bas; endothelial) side alone or on both sides simultaneously. TER remained unaltered and stable after LPS ap exposure for 48 h. LPS bas exposure caused a barrier disruption to ca. 70% (compared to the untreated control uc). Stimulation with the cytokine mixture 3C bas decreased the TER to a similar extent, as well as with co-stimulation with LPS bas. Co-stimulation with LPS ap, however, did not cause a decrease in TER after 3C bas treatment. The permeability of NaFlu increased slightly but not significantly after stimulation with LPS ap, LPS bas and LPS ap / 3C bas. Only 3C bas with or without LPS bas increased P_app_ significantly.

Apical stimulation with LPS (LPS ap) did not cause any change in the release of IL-8, sICAM-1 or sE-selectin (Figure 3A–C) on the apical or on the basolateral side. Basolateral stimulation with LPS (LPS bas) only resulted in a significant basolateral increase of IL-8 but not sICAM-1 or sE-selectin. The addition of 3C bas resulted in significant IL-8 production in the basolateral compartment, but not in the apical compartment. However, 3C in combination with LPS (LPS ap /3C bas, 3C + LPS bas) resulted in a significantly increased IL-8 release in both apical and basolateral compartments. Furthermore, 3C stimulation, together with LPS ap and bas, resulted in a significant and comparable increase of sICAM on both sides and sE-selectin only on the endothelial side. No sE-selectin could be detected in the upper well under any of the stimulation conditions. Stimulation with 3C on the endothelial side was used as the model to trigger inflammation for the following studies with inverted coculture ISO-HAS-1/Caco-2 (CCinv).

Figure 4 shows the inflammatory response (sICAM-1 and sE-selectin) of the inverted coculture ISO-HAS-1/Caco-2 (CCinv) upon stimulation of the endothelial side (in this case the apical side) with 3C in combination with different surface-modified amorphous silica nanoparticles (aSNPs). Interestingly, for all aSNPs with a diameter of 70 nm (-plain, -NH2, -COOH), a significant decrease of sICAM-1 and sE-selectin could be detected after stimulation with the cytokine mixture 3C (compared to 3C stimulation without aSNPs). For the aSNP (-plain) with a diameter of 30 nm, no obvious alteration of sICAM-1 and sE-selectin release occurred on both sides. In the basolateral compartment (epithelial side), no altered sICAM-1 response occurred after aSNP exposure on the endothelial side, compared to uc and 3C without aSNPs, respectively. For sE-selectin, however, a reduction could be observed on both sides after endothelial exposure of the various aSNPs in combination with 3C (compared to 3C *w*/*o* aSNPs).

Figure 5 shows the ELISA measurement of sICAM-1 (Figure 5A) and sE-selectin (Figure 5B) of the same supernatants as used for the ELISA shown in Figure 4 after the supernatants were used for the total exosome isolation procedure. For this, supernatants of all three independent experiments (with *n* = 3) were pooled, exosomes were isolated, resuspended in PBS (phosphate buffered saline), and then the exosome suspensions were analyzed for the presence of sICAM-1 and sE-selectin by ELISA. The values under inflammatory conditions (exposed to the cytokine mixture 3C, CCinv, endothelial side) are depicted with simultaneously exposure to the aSNPs with different properties (30 nm –plain and 70 nm –plain, -COOH, –NH2 as well without aSNPs: *w*/*o* aSNPs). Only the endothelial side (apical compartment) was used for comparison to Figure 4. Figure 5A shows a similar sICAM-1 response compared to Figure 4B after exposure to the cytokine mixture 3C compared to the untreated control uc. Exposure to 30 nm aSNP in combination with 3C showed no alteration compared to the 3C stimulation without aSNPs (*w*/*o* aSNPs). All 70 nm aSNPs in combination with 3C, however, caused a similar reduction in sICAM release compared to the respective stimulation in Figure 4B on the endothelial side of the membrane. A similar response pattern could be observed for sE-selectin in Figure 5B compared to Figure 4C). All aSNPs (30 nm and all 70 nm aSNPs) caused a reduction of the sE-selectin signal in combination with the 3C stimulation as it is detected in the respective stimulation in Figure 4C.

Thus, the increased sICAM-1/sE-selectin ELISA readings, which were obtained in the 3C inflamed coculture (CCinv) on the endothelial side, may be due to an increased release of ICAM-1/E-selectin positive exosomes/microvesicles, instead of, or in addition to, the soluble forms of ICAM-1 and E-selectin. Furthermore, the 3C-induced increase of exosome release was reduced upon simultaneous treatment with aSNP.

As a control, the highest standard concentration of recombinant sICAM and sE-selectin (St) (sICAM-1: 2000 pg/mL and for sE-selectin: 6000 pg/mL) diluted in cell culture media ECGM with 15% FCS underwent the same total exosome isolation procedure (St Exo). The results show that these were negative and exhibited similar values to the readings for the cell culture media (ccm), compared to the standards prior to exosome isolation (St).

In Figure 6, the barrier properties based on TER (Transepithelial resistance, (% of t0), t0: time point prior to stimulation of each well respectively) and P_app_ (apparent permeability coefficient of sodium fluorescein: NaFlu) of the inverted coculture CCinv are shown. None of the different-sized or surface-modified aSNPs caused significant alterations to the barrier properties in the control cells or cells exposed to the inflammatory conditions CCinc (3C).

## 3. Discussion

This study assessed the immuno-responsivity of an entero-capillary barrier in vitro model mimicking a “healthy” and a “pathophysiological” state as it would occur for inflammatory bowel disease (IBD). To evaluate the role of the intestinal microvasculature, inflammation was triggered via different stimuli on both barrier sides respectively. Simultaneously, the impact of the addition of different amorphous silica nanoparticles (aSNPs), which are being considered as putative drug vehicles, was monitored during ongoing inflammation. In addition to the barrier integrity (TER and P_app_), the inflammatory profile (release of IL-8, sICAM and sE-selectin) was examined specifically with a special emphasis on sICAM and sE-selectin as putative markers for exosomes. The release of sICAM and sE-selectin appears to be altered during silica nanoparticle treatment.

Distinct stimulation protocols were first evaluated in this study to trigger inflammation, which, in turn, led to a disruption of the barrier integrity. Stimulation of the endothelial side with a cytokine mixture, 3C (TNF-α, Interferon-γ and Interleukin 1-β) was used to simulate a systemic or vascular inflammation in vivo. A severe barrier disruption and an increased expression of all three inflammatory mediators (IL-8, sICAM and sE-selectin) was observed. These conditions were utilized in all subsequent studies. As has been shown in several clinical and animal studies [17,18,19,20], IBD is accompanied by a drastic disruption of the intestinal barrier, which is often caused by a systemic immune activation coming from the endothelial side [21,22]. Thus, the in vitro model closely mimics what occurs in IBD in vivo. To evaluate the relevance of endothelial cells in the in vitro model after silica nanoparticle treatment during inflammation the coculture with the Caco-2/ISO-HAS-1 was inverted so that the ISO-HAS-1 were present on the top and the Caco-2 cells on the bottom side of the transwell filter. In this way, compounds of interest could be applied directly to the endothelial cells.

### 3.1. aSNP Exposure to the Endothelium Decreases sICAM and sE-Selectin Levels on both Sides of the Inflamed Coculture Model

A reduction of sICAM-1 release was observed on the endothelial side of the coculture after exposure to each of the different modified 70 nm-aSNPs to the 3C-stimulated endothelium (compared to 3C stimulation without aSNPs). Since no significant sICAM reduction was observed with aSNP with the 30 nm, it appeared that effects of NPs were size-dependent. A reduction of sE-selectin release was observed on both sides of the coculture after exposure to the three different 70 nm-aSNPs to the 3C-stimulated endothelium (compared to 3C stimulation without aSNPs). aSNPs with a diameter of 70 nm decreased sICAM to a significantly greater extent compared to aSNP with a 30 nm diameter, which was not significantly changed. A clear difference between different surface modifications of the aSNP could not be seen; however, the effects with aSNP-COOH appeared to be slightly higher.

Many previous studies have shown increased levels of the soluble form of both ICAM-1 and E-selectin released into the systemic circulation that correlate with the activity and severity of several diseases such as rheumatoid arthritis [23,24], Grave’s disease [25,26] and systemic vasculitis [27]. High sICAM-1 levels are also associated with cardiovascular risk factors such as hypertension, smoking and alcohol abuse [28,29]. As has been previously discussed in Kasper et. al. [15], high levels of sICAM-1 and sE-selectin in the blood circulation in active colitis ulcerosa (UC) and Crohn’s disease serve as an indicator of disease severity [11,30,31]. The results of the present study further support the implication of sICAM-1 and sE-selectin release in an in vitro intestinal cell culture model under inflammatory conditions as indicators of disease. These two factors are present in all cases of diseases associated with vascular wall inflammation [32], and these, in turn, may affect the functional and protective characteristics of biological barriers.

### 3.2. The Endothelial Release of ICAM and E-Selectin-Bearing Exosomes/Microvesicles Is Increased during Endothelial Inflammation and Reduced during Endothelial aSNP Treatment

Elevated ICAM and E-selectin levels in the supernatants after apical (endothelial) 3C stimulation were detected via ELISA and most likely represent at least partially ICAM/E-selectin bearing exosomes/microvesicles. Microvesicles directly bud from the plasma membrane, whereas exosomes are secreted via an exocytotic mechanism from multivesicular bodies (MVBs) of the late endosome [33,34,35]. Once released into the extracellular environment they contribute to intercellular communication via their specific composition of surface molecules as well as their specific cargo molecules. Exosomes from hypoxic endothelial cells are reported to influence e.g., extra cellular matrix remodeling [36]. Exosomes released from different cell types promote angiogenesis under certain circumstances [37,38,39,40]. They are also considered as a promising application for cancer immunotherapy, since they may affect regulatory functions in the immune system [41,42,43]. Lee and coworkers found that exosomes, which bear ICAM-1 but not E-selectin on their membrane display an anti-inflammatory function, since they efficiently inhibit leukocyte adhesion to activated endothelial cells. In contrast, the soluble form of sICAM-1 [44] did not show this effect. It has a low affinity for its ligand LFA-1, and thus, shows a low immune response inhibition [45]. Dignat-George and Boulanger have shown that activated endothelial cells release microvesicles (ca. 100 nm to 1 µm in diameter) that bear ICAM-1 and E-selectin on their surface. These endothelial microparticles (EMP) among others may serve as potential biomarkers for vascular injury. The EMPs exhibit many functions with anti- as well as pro-inflammatory potential, affecting endothelial function, such as angiogenesis and vascular homeostasis [12].

In this study, the endothelial cells under “activated” culture conditions (stimulation with cytokine mixture C3) in the presence of aSNPs exhibited a decreased level of ICAM-1 positive microvesicles or exosomes in the supernatant. According to the above-mentioned studies, this would indicate an impaired immune response by an activated endothelium under ongoing “vascular inflammation” (C3 stimulation). No distinction was made in the present study between smaller exosomes or larger microvesicles. These need to be evaluated in future studies to determine exosomal size and surface markers as well as their cargo. Furthermore, additional studies should focus on a more detailed characterization that includes imaging via transmission electron microscopy and the evaluation of exosomal markers e.g., CD63, CD9, ALIX or CD86, which are recommended as the minimal criteria by which to define exosomes according to the MISEV-2018 guidelines [46].

Recent studies have demonstrated that aSNPs (specifically Sicastar, as used in the present study), were incorporated into late endosomal structures after incubation with lung epithelial (NCI H441) and endothelial cells (ISO-HAS-1) [47,48]. Thus, it appears that internalized aSNPs present in the late endosomal structures may compromise the plasticity of the endosomal structure or may decelerate processes taking place in the endosomal sorting station, in which lysosomal degradation, recycling to the cell surface or exosomal formation and secretion is coordinated. Thus, although nanoparticles may not show cytotoxicity, they may hamper cellular processes in general, and thereby, compromise a proper cellular response. This could result in a changed cellular communication which would occur physiologically upon contact with inflammatory stimuli.

The present model could be modified to more closely mimic the in vivo conditions of the iMV in vitro through the addition of other tissue-relevant cell-types such as monocytes/macrophages or dendritic cells [1,2,3,4], intraepithelial lymphocytes [5], fibroblasts [6], enteric glial cells [7,8] or intestinal pericytes [49]. These cell types have a critical influence on the iMV. Furthermore, using primary cells instead of cell lines may provide a model more closely resembling the in vivo tissue. Cells lines may always lead to misrepresenting results due to their altered features. Additionally, it is known that nanoparticles themselves have the ability to cross the endothelial barrier. Recent studies by Ho and coworkers described methods to investigate the extravasation of nanoparticles [50]. aSNP uptake and translocation experiments have been described for ISO-HAS-1 using the lung equivalent in vitro model of the alveolar-capillary barrier [48]. In the transwell model, aSNPs were able to cross the semi-permeable transwell filter membrane with a pore size of 400 nm. However, low pore size may decelerate aSNP translocation. Furthermore, cellular uptake experiments indicated a colocalization of internalized aSNPs with flotillin bearing endosomes [47,48]. Thus, additional studies are necessary to obtain a better understanding of the intracellular uptake and translocation of aSNPs in the gut blood barrier in vitro.

## 4. Materials and Methods

Nanoparticles: Sicastar^®^-Red are amorphous silica nanoparticles (aSNP, specification: spherical, unporous, rho = 2 mg/cm^3^) in aqueous dispersion with a nominal diameter of 30 or 70 nm, respectively (micromod Partikeltechnologie GmbH, Rostock, Germany (www.micromod.de)). Sicastar Red is fluorescently labelled with Rhodamin B (Ex: 569 nm, Em: 585 nm), which is covalently attached to the SiO_2_ matrix. For aSNPs with a diameter of 70 nm several different surface modifications were investigated (Si–OH/Si–O–(-plain); carboxy (–COOH) and amine (–NH2)). For the 30 nm aSNP the unmodified (-plain) was used. More detailed information concerning the characterization of these aSNPs, their size and dispersion behavior in cell culture media have been previously described [47,48,51].

Cell culture: Caco-2 (human epithelial colorectal adenocarcinoma cell line) was purchased from ATCC^®^ (HTB-37™). The cells were cultured in DMEM (High Glucose, Gibco, 41965-039, life technologies, Paisley, UK) supplemented with 10% fetal calf serum, 1% penicillin/streptomycin (100 U/100 µg/mL), 1% Glutamax I (Gibco, 35050-038, life technologies, Paisley, UK) and 1% NEAA (non-essential amino acids, Gibco, 11140, life technologies, Paisley, UK) at 37 °C, 5% CO_2_. ISO-HAS-1 (human microvascular endothelial cell line [52,53]) was cultured in ECGM (Endothelial Cell Growth Medium MV, PromoCell, Heidelberg, Germany) supplemented with 15% FCS, Pen/Strep (100 U/100 µg/mL), 0.2% bFGF (basic fibroblast growth factor, Sigma, F0291, Darmstadt, Germany) and 0.02% Na-Heparin, PromoCell, Heidelberg, Germany. ISO-HAS-1 and Caco-2 were cultivated at 37 °C, 5% CO_2_ and passaged every third day at a dilution of 1:3 and 1:8, respectively.

The coculture model of the intestinal barrier Caco-2/ISO-HAS-1: The coculture technique was previously described for the lung air-blood barrier H441/ISO-HAS-1 [48,54] and for the intestinal barrier Caco-2/ISO-HAS-1 [15]. Briefly, HTS 24-Transwell^®^ filters (polycarbonate, 0.4 µm pore size; Costar, Wiesbaden, Germany) were coated with rat tail collagen type-I (12.12 µg/cm^2^, BD Biosciences, Heidelberg, Germany) on both sides. The transwell plate was inverted and the ISO-HAS-1 cells were seeded on the lower side (1 × 10^4^ cells/well). After 2 h at 37 °C and 5% CO_2_, to allow for the adhesion of cells, the transwells were inverted and the Caco-2 cells were seeded into the upper chamber at 6.4 × 10^3^ cells per well. Cells were cultured for 21 days (upper well: 200 µL Caco-2 cell culture medium; lower well: 1 mL ISO-HAS-1 cell culture medium). Cell culture medium was replaced two times per week for the Caco-2 and 3 times for ISO-HAS-1 cells. In addition, the coculture was set up in reverse. Here the endothelial cells ISO-HAS-1 were seeded on the upper side and the Caco-2 were seeded on the lower side of the membrane.

Immunofluorescence (IF) staining: the IF staining technique was previously described [15]. Briefly, Caco-2 cells in coculture with ISO-HAS-1 on transwell membranes were fixed with PFA 3.7% (Paraformaldehyde) in CS-Buffer (0.1 M Pipes, 1 mM EGTA, 4% polyethyleneglycol 8000, 0.1 M NaOH) for 20 min RT. Membranes were then rinsed 3× in PBS and permeabilized with 2% Triton X100 in PBS for 20 min at RT. After washing 3× with PBS, the Caco-2 side of the membrane was incubated with primary antibodies (diluted in PBSA 1%) against ZO-1 (zonula occludens-1, 1:200, Zymed 61-7300, Thermo Fisher Scientific, Dreieich, Germany) and β-catenin (1:400, BD, 610154, Heidelberg, Germany) and the ISO-HAS-1 side with antibodies against CD31 (1:50, Dako, M 0823, Santa Clara CA, USA). Membranes were washed 3× with PBS and then the secondary antibodies (1:1000 in PBSA 1%; Alexa 488, anti-mouse, Invitrogen A11029 and Alexa 546, anti-rabbit, Invitrogen A11010, Thermo Fisher Scientific, Dreieich, Germany) were added and incubated for 1 h at RT. Both sides (epithelial and endothelial side) were stained simultaneously by adding AB solution to the upper and lower well. After washing 3× with PBS, cells were stained with Hoechst 33342 (1:10,000 in PBS, Thermo Fisher Scientific, Dreieich, Germany) for 5 min RT and washed three times. Finally, transwell filters were removed by cutting around the edge of the well and were mounted with Fluoromount-G™ (Southern Biotech, Birmingham, AL, USA). Images were obtained using a fluorescence microscope (DeltaVision, Applied Precision, Issaquah, WA, USA).

Stimulation of the coculture with inflammatory mediators: The apical (upper compartment, epithelial side) or basolateral (lower compartment, endothelial side) side of the coculture CC was treated with LPS (lipopolysaccharide, 1 µg/mL) and the basolateral side with a cytokine mixture 3C (IL-1β (100 U/mL), TNF-α (600 U/mL) or IFN-γ (200 U/mL)) alone and in combination with LPS for 48 h. All stimulants were applied to the cells on the transwells in a final working volume of 200 µL/well (20 µL stimulant solution + 180 µL cell culture medium). The inverted coculture CCinv model ISO-HAS-1/Caco-2 was used to examine the effects of aSNPs. In this case, 3C and aSNPs and a combination of both were added to the apical side (upper compartment, endothelial side, lower compartment, Caco-2) under the conditions described above. The aSNP suspensions were applied to the upper compartment (containing the endothelial cells) at a concentration of 100 µg/mL.

Transepithelial electrical resistance: The barrier integrity was determined via transepithelial electrical resistance (TER) as previously described [15]. TER was measured before and after the 48 h incubation period with the stimulants using an EVOM volt-ohmmeter (World Precision Instruments, Berlin, Germany) equipped with a STX-2 chopstick electrode. An empty well (without cells) was utilized as blank (approximately 110 Ω). Barrier resistance readings (Ω) were obtained for each well individually. After subtracting the blank, the readings were then multiplied by the membrane area (0.33 cm^2^) to give Ω*cm^2^. The TER value of each single well before the stimulation t0 was set as 100% to normalize the value of the same well after 24 h.

Transport assay: After 48 h incubation with the various stimulants, 10 µg/mL sodium fluorescein (NaFlu) was applied to the upper compartment of the transwell as described previously [15]. After 3 h, 50 µL of the cell culture medium from the lower compartment was removed to a 96-well plate containing 200 µL 0.4 mM NaOH. Fluorescence (EX 488) was measured in a fluorescence plate reader. The apparent permeability coefficient (P_app_) was determined via the equation: P_app_ (cm/s) = (1/(A*C0))*(dQ/dt) with A= filter surface (0.33 cm^2^), C0: Concentration of NaFlu in the donor solution (10 µg/mL), dQ/dt (paracellular flux) = amount of NaFlu (µg/s), which is transported across the cellular layer within 3 h.

Inflammatory responses: The supernatants were examined for the presence of IL-8, soluble (s)ICAM-1 (intercellular adhesion molecule-1) and soluble (s)E-selectin released by the cells via ELISA (enzyme linked immunosorbent assay: DuoSet R&D, DY208, DY720, DY724, Minneapolis, MN, USA) according to the manufacturer’s instructions.

Total Exosome Isolation: The supernatants from endothelial cells from three independent experiments (with *n* =3), which were examined for inflammatory mediators via ELISA as described above, were then pooled and exosomes were isolated via the Total Exosome Isolation Kit (Invitrogen, 4478359, TF, Dreieich, Germany) according to the manufacturer’s instructions. After isolation, the purified exosome suspensions were examined via ELISA for sICAM-1 and sE-selectin. In addition, the standard recombinant sICAM-1 and sE-selectin proteins supplied in the ELISA kits were subjected to the exosome isolation procedure in order to determine if the soluble form of the two proteins was co-precipitated during the exosome isolation procedure and detected in the ELISA of the exosome suspensions.

Statistical analysis: From several independent measurements, means and standard errors were calculated. Data are shown as mean ± SEM from at least three separate experiments with *n* = 3. Testing for significant differences between means was carried out using one-way ANOVA and Dunnett’s Multiple Comparison test at a probability of error of 5% (*), 1% (**), 0.1% (***) and 0.01% (****). Graphs and statistical analysis were conducted via GraphPad Prism 5, San Diego, CA, USA)

Graphical abstract: schematic drawings of cells were partly performed with Biomedical PPT Toolkit Suite, Motifolio, Inc., Ellicott Citty, MD, USA.

## 5. Conclusions

In this study, an in vitro model of the entero-capillary barrier was developed that exhibits barrier disruption and inflammatory behavior mimicking the pathophysiological in vivo conditions associated with IBD. Surprisingly, increased levels of the IBD-biomarkers sICAM and sE-selectin correlated with an increase of ICAM-1 and E-selectin bearing exosomes/microvesicles. This elevation could be significantly reduced after treatment with amorphous silica nanoparticles, which are being considered as prospective drug-delivery vehicles. These findings suggest altered exosomal/microvesicular communication within the endothelial lining after exposure to silica nanoparticles. Whether this scenario compromises effective systemic immune responses remains unclear, and needs to be evaluated in future studies to reveal the nature of these vesicles and their cargo.

## Figures and Tables

**Figure 1 ijms-20-03301-f001:**
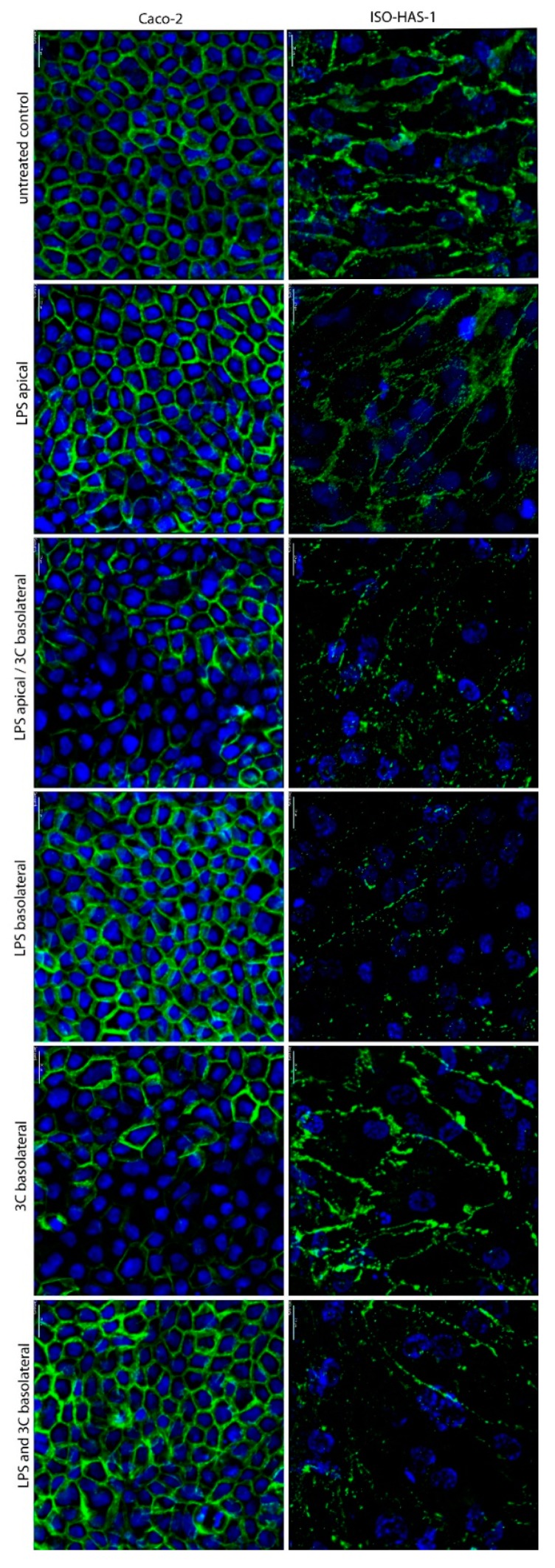
Immunofluorescence staining of the coculture CC (Caco-2/ISO-HAS-1) after stimulation with different pro-inflammatory mediators for 48 h. apical: stimulation in the upper well (Caco-2), basolateral: stimulation in the lower well (ISO-HAS-1); LPS: lipopolysaccharide (1 µg/mL); 3C: cytokine mixture ((IL-1β (100 U/mL), TNF-α (600 U/mL) and IFN-γ (200 U/mL)); green signal: immunofluorescence of β-Catenin for both cell types Caco-2 (**left** column) and ISO-HAS-1 (**right** column); blue signal: nuclei stained with Hoechst 33342; scale bar: 20 µm.

**Figure 2 ijms-20-03301-f002:**
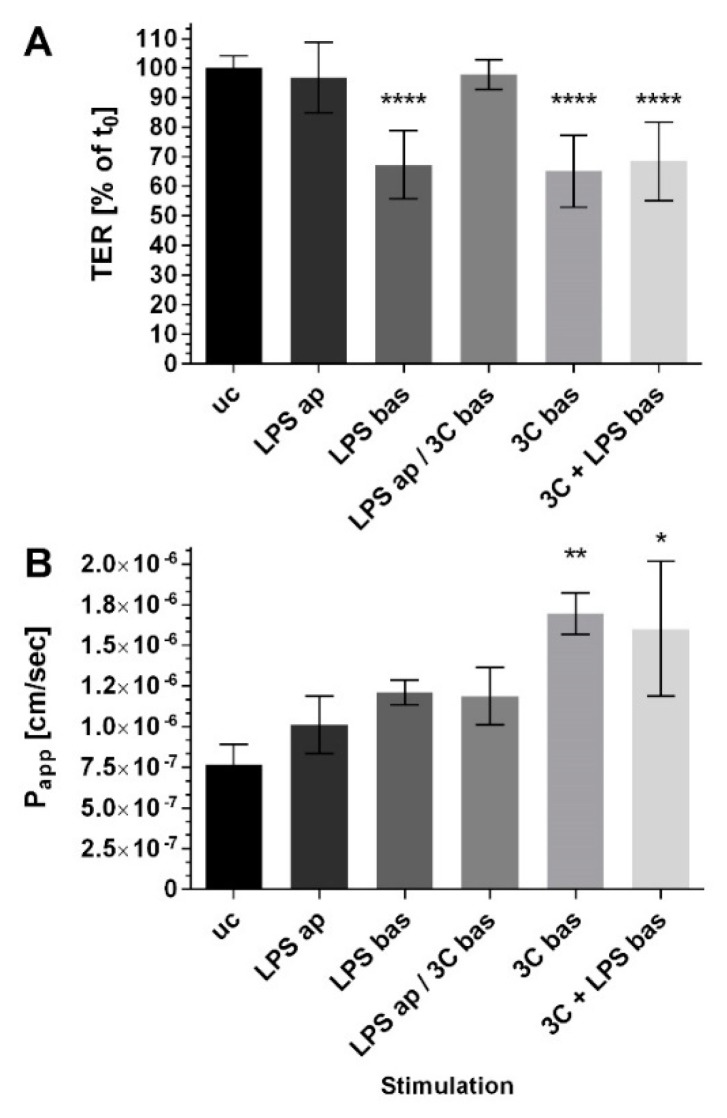
(**A**) Transepithelial electrical resistance measurements TER (% of t0), t0: time point prior to stimulation of each well, respectively; (**B**) transport assay using NaFlu, P_app_: the apparent permeability coefficient (cm/s) of the coculture CC (Caco-2/ISO-HAS-1) after stimulation with different pro-inflammatory mediators for 48 h. ap: apical, stimulation in the upper well (Caco-2), bas: basolateral, stimulation in the lower well (ISO-HAS-1); LPS: lipopolysaccharide (1 µg/mL); 3C: cytokine mixture ((IL-1β (100 U/mL), TNF-α (600 U/mL) and IFN-γ (200 U/mL)); data are depicted as means ± SE of three independent experiments with *n* = 3. For statistical analysis one-way ANOVA with Dunnett’s Multiple Comparison test was conducted. * *p* < 0.05, ** *p* < 0.01, and **** *p* < 0.0001.

**Figure 3 ijms-20-03301-f003:**
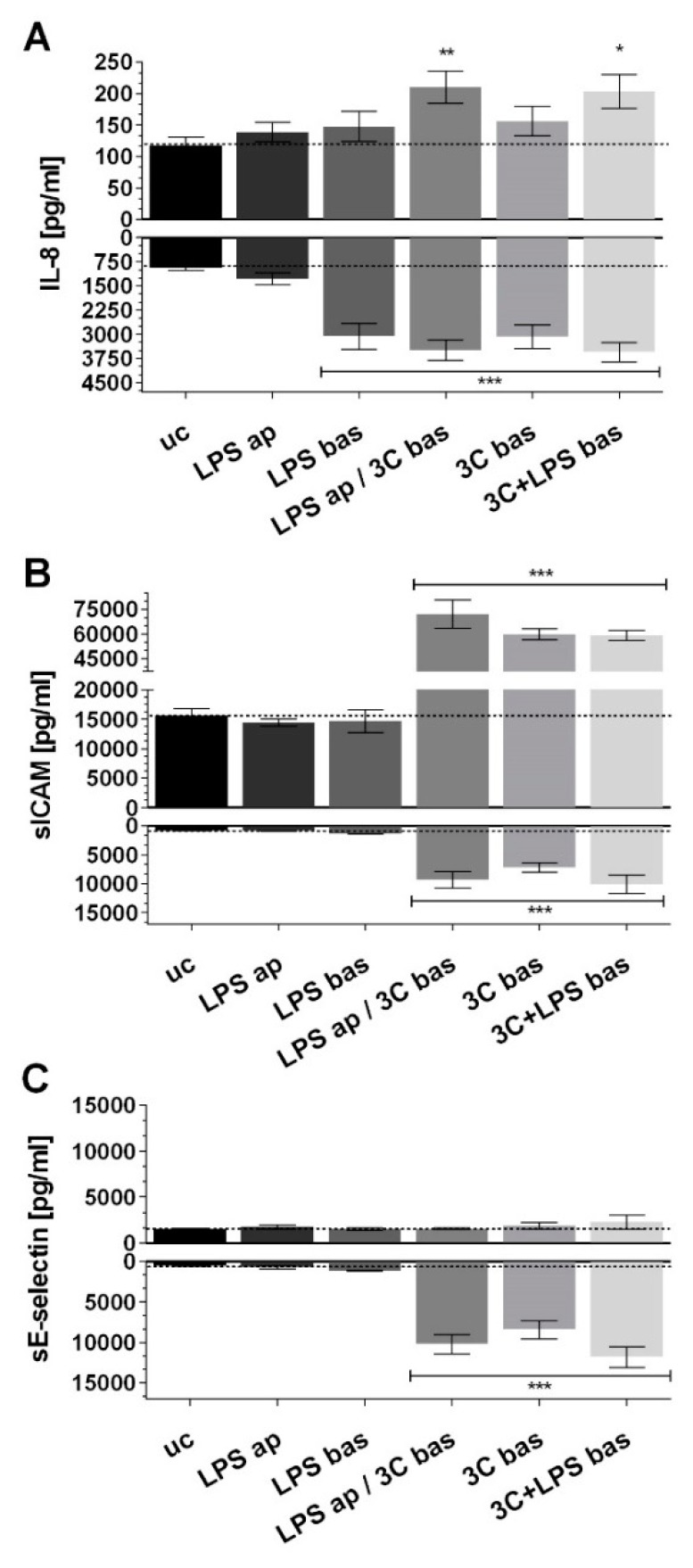
ELISA (enzyme linked immunosorbent assay) of (**A**) IL-8; (**B**) sICAM-1 and (**C**) sE-selectin (pg/mL) after stimulation of the coculture CC with different pro-inflammatory mediators for 48 h. ap: apical, stimulation in the upper well (Caco-2), bas: basolateral, stimulation in the lower well (ISO-HAS-1); LPS: lipopolysaccharide (1 µg/mL); 3C: cytokine mixture ((IL-1β (100 U/mL), TNF-α (600 U/mL) and IFN-γ (200 U/mL)); release of inflammatory mediators in the upper (upward columns) compartment (apical) and lower (downward columns, basolateral); data are depicted as means ± SE of three independent experiments with *n* = 3. For statistical analysis one-way ANOVA with Dunnett’s Multiple Comparison test was conducted. * *p* < 0.05, ** *p* < 0.01, *** *p* < 0.001.

**Figure 4 ijms-20-03301-f004:**
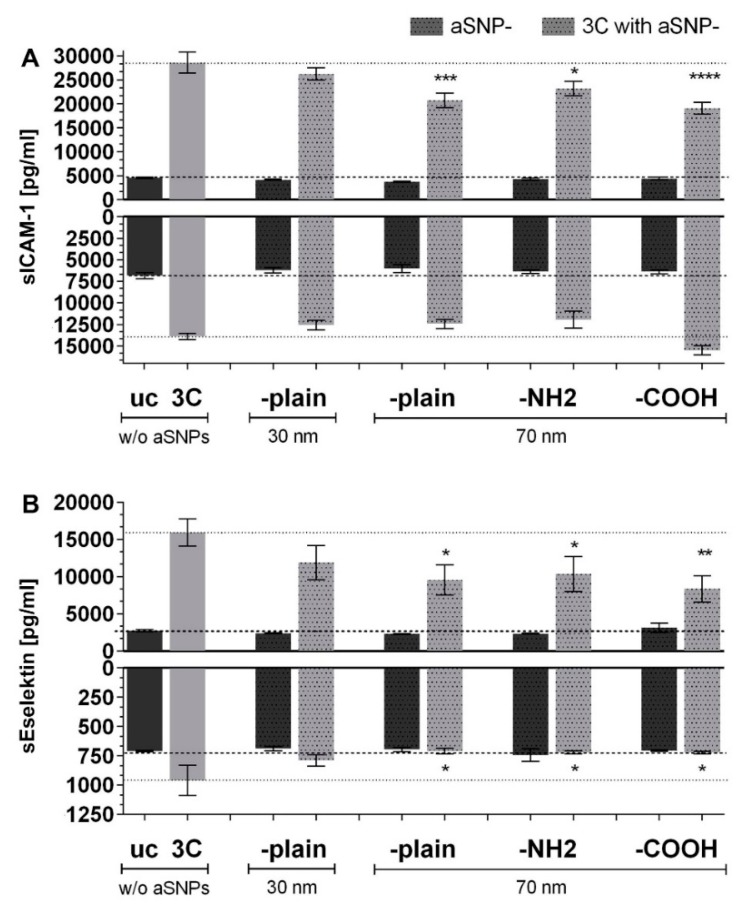
ELISA (enzyme linked immunosorbent assay) of (**A**) sICAM-1 and (**B**) sE-selectin (pg/mL) after stimulation of the inverted coculure CCinc (ISO-HAS-1/Caco-2) with aSNPs and 3C (cytokine mixture (IL-1β (100 U/mL), TNF-α (600 U/mL) and IFN-γ (200 U/mL)) for 48 h; uc: untreated control (without aSNP and 3C); aSNPs: -plain, 30 nm: Sicastar Red with a diameter of 30 nm, without surface modifications; -plain, -NH2, -COOH, 70 nm: Sicastar Red with a diameter of 70 nm and different surface modifications. All aSNPs were applied at a concentration of 100 µg/mL. Data are depicted as means ± SE of three independent experiments with *n* = 3. For statistical analysis one-way ANOVA with Dunnett’s Multiple Comparison test was conducted. * *p* < 0.05, ** *p* < 0.01, *** *p* < 0.001 and **** *p* < 0.0001; #: comparison within the group without 3C (dark grey columns, no significance observed); *: comparison within the group with 3C (light grey columns).

**Figure 5 ijms-20-03301-f005:**
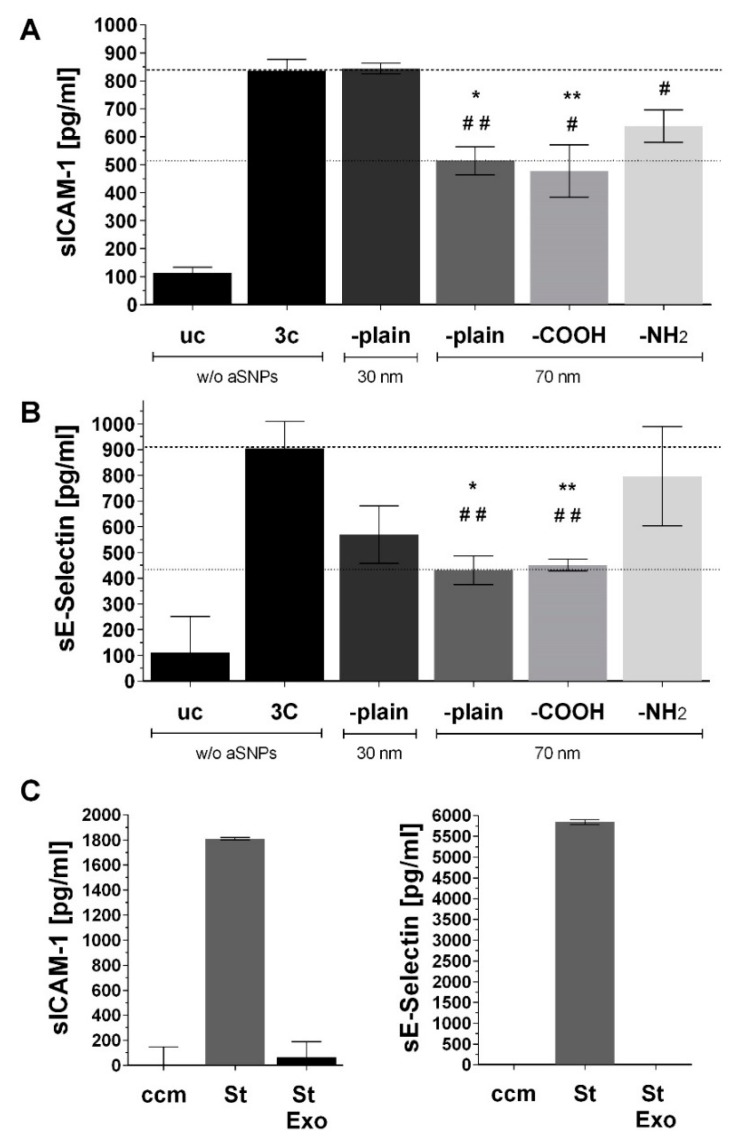
ELISA (enzyme linked immunosorbent assay) of (**A**) sICAM-1 and (**B**) sE-selectin (pg/mL) of the same supernatants (only endothelial side) as used for the ELISA in Figure 4 after total exosome isolation. Stimulation of the inverted coculture CCinc (ISO-HAS-1/Caco-2) with aSNPs and 3C (cytokine mixture ((IL-1β (100 U/mL), TNF-α (600 U/mL) and IFN-γ (200 U/mL)) for 48 h; uc: untreated control (without aSNP and 3C); aSNPs: -plain, 30 nm: Sicastar Red with a diameter of 30 nm, without surface modifications; -plain, -NH2, -COOH, 70 nm: Sicastar Red with a diameter of 70 nm and different surface modifications. All aSNPs were applied in a concentration of 100 µg/mL. data are depicted as means ± SE of three pooled experiments with *n* = 3. (**C**) ELISA of control groups St: recombinant protein (sICAM-1: 2000 pg/mL) and sE-selectin: 6000 pg/mL); ccm: cell culture medium; St Exo: recombinant protein, which underwent total exosome isolation procedure. For statistical analysis one-way ANOVA with Dunnett’s Multiple Comparison test was conducted. * and # *p* < 0.05, ** or ## *p* < 0.01, (#: comparison to uc; *: comparison with 3C).

**Figure 6 ijms-20-03301-f006:**
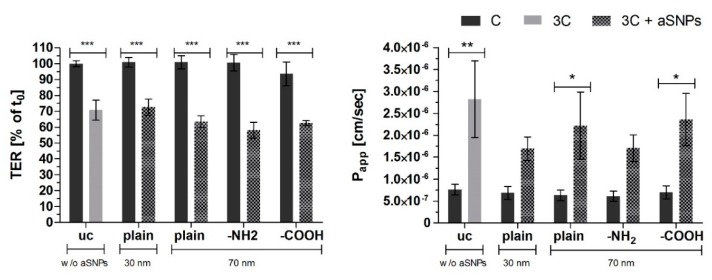
TER: Transepithelial electrical resistance measurements (% of t0), t0: time point prior to stimulation of each well, respectively; P_app_: the apparent permeability coefficient (cm/s)) of NaFlu through the coculture CC (Caco-2/ISO-HAS-1) after stimulation of the inverted coculure CCinc (ISO-HAS-1/Caco-2) with aSNPs and 3C (cytokine mixture ((IL-1β (100 U/mL), TNF-α (600 U/mL) and IFN-γ (200 U/mL)) for 48 h; uc: untreated control (without aSNP and 3C); aSNPs: -plain, 30 nm: Sicastar Red with a diameter of 30 nm, without surface modifications; -plain, -NH2, -COOH, 70 nm: Sicastar Red with a diameter of 70 nm and different surface modifications. All aSNPs were applied in a concentration of 100 µg/mL. Data are depicted as means ± SE of three independent experiments with *n* = 3. For statistical analysis one-way ANOVA with Dunnett’s Multiple Comparison test was conducted. * *p* < 0.05, ** *p* < 0.01, *** *p* < 0.001.

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
