# Peer review of "In Vitro Entero-Capillary Barrier Exhibits Altered Inflammatory and Exosomal Communication Pattern after Exposure to Silica Nanoparticles"

_ijms, 2019, doi:10.3390/ijms20133301_

Reviewer 1 Report

The manuscript "An in vitro entero-capillary barrier exhibits an altered inflammatory and exosomal communication pattern after exposure to silica nanoparticles" by J. Y. Kasper et al. is carefully prepared, nicely structured and well written. The authors use appropriate statistical assessment and the data are convincing and clearly presented. 

There are only a few minor rather technical suggestions: 

1. The main text clearly refers to inflammatory mimicking conditions while the abstract is missing this distinction - in fact I believe line 26/27 (ongoing inflammation/inflamed endothelium) refers to the in vitro model. In contrast line 29 (inflamed endothelium) seems to refer to in-vivo.

line 63 - one space too much before the dot.

line 72 space too much after (...)response. 

line 183, one space too much ? (not clear) 

line 360 'as described previously' is missing the reference.

A few thoughts while reading the manuscript: 

(the authors might choose to include a few sentences in the manuscript but not required) 

The developed model seems to represent the in-vivo environment reasonably well. However, without any doubt the in-vivo situation during inflammation is much more complex involving further cell types such as pericytes, macrophages, neutrophils etc. 

Gut Pericytes likely play an important role during  inflammation: 

Scientific Reports volume 7, Article number: 39848 (2017) Park and co-workers

Including pericytes is challenging though, since they tend to form cords with 2D endothelial layers, disrupting any barrier function.  Additional, different in-vitro assays would need to be used maybe adaption of the assays reported last year year in this journal might work.

 Int. J. Mol. Sci. 201819(10), 2913; https://doi.org/10.3390/ijms19102913

It might be good to highlight the limitation of the current model and how more complex in-vitro systems could be generated in the future. I do not think that it would diminish the value of the present study in any way nor would I expect the general nature of the data to change in a more complex system. 

Nanoparticles themselves have the ability to cross the endothelial barrier

Scientific Reports 7, Article number: 707 (2017) (Ho, Adriani, et al)

It would be interesting to learn whether the authors observed trans-barrier (migration/transport) across both cell layers (endothelial/epithelial) or maybe the nanoparticles accumulated in between the two cell layer. 

However, no additional experiments need to be performed, if the authors happen to have data it would certainly not harm to reference inside the manuscript to this observations.

Author Response

Dear Prof. Dr. Jellinger, Dear Ms. Wu, dear reviewer,

thank you for your quick response regarding our manuscript submitted to your special issue entitled „Endothelial Dysfunction“ of the „International Journal of Molecular Sciences“ with the title: „An in vitro entero-capillary barrier exhibits an altered inflammatory and exosomal communication pattern after exposure to silica nanoparticles“. The coautors are Jennifer Y. Kasper, M. Iris Hermanns, Annette Kraegeloh, W. Roth, C. James Kirkpatrick, Ronald E. Unger.

Attached you will find our responses to the reviewer´s comments.

Comment 1:

„1. The main text clearly refers to inflammatory mimicking conditions while the abstract is missing this distinction - in fact I believe line 26/27 (ongoing inflammation/inflamed endothelium) refers to the in vitro model. In contrast line 29 (inflamed endothelium) seems to refer to in-vivo.“

-       A clear description concerning the inflammatory mimicking conditions of the model has been added to the abstract.  Line 26/27 and line 29 were rewritten for a better understanding as follows:

Abstract: The intestinal microvasculature (iMV) plays multiple pathogenic roles during chronic inflammatory bowel disease (IBD). The iMV acts as a second line of defense and is among others crucial for the innate immunity in the gut. It is also the therapeutic location in IBD targeting aggravated leukocyte adhesion processes involving ICAM-1 and E-selectin. Specific targeting is stressed via nanoparticulate drug vehicles. Evaluating the iMV in enterocyte barrier models in vitro could shed information on inflammation and barrier-integrity processes during IBD. Therefore, we generated a barrier model by combining the enterocyte cell line Caco-2 with the microvascular endothelial cell line ISO-HAS-1 on opposite sides of a transwell filter-membrane under culture conditions mimicking physiological and inflamed conditions resembling IBD. The IBD model achieved a significant barrier-disruption demonstrated via transepithelial-electrical resistance (TER), permeability-coefficient (Papp) and increase of sICAM sE-selectin and IL-8. In addition, the impact of a prospective model drug-vehicle (silica nanoparticles, aSNP) on ongoing inflammation was examined. A decrease of sICAM/sE-selectin was observed after aSNP-exposure to the inflamed endothelium. These findings correlated with a decreased secretion of ICAM/E-selectin bearing exosomes/microvesicles, which we evaluated via ELISA. Our findings indicate that aSNP treatment of the inflamed endothelium during IBD may hamper exosomal/microvesicular systemic communication.

Comment 2:

„line 63 - one space too much before the dot.

line 72 space too much after (...)response. 

line 183, one space too much ? (not clear) 

line 360 'as described previously' is missing the reference. â€ž

-       The above errors have been corrected.

Comment 3:

„A few thoughts while reading the manuscript: 

(the authors might choose to include a few sentences in the manuscript but not required) 

The developed model seems to represent the in-vivo environment reasonably well. However, without any doubt the in-vivo situation during inflammation is much more complex involving further cell types such as pericytes, macrophages, neutrophils etc. 

Gut Pericytes likely play an important role during inflammation: 

Scientific Reports volume 7, Article number: 39848 (2017) Park and co-workers

Including pericytes is challenging though, since they tend to form cords with 2D endothelial layers, disrupting any barrier function.  Additional, different in-vitro assays would need to be used maybe adaption of the assays reported last year year in this journal might work.

 Int. J. Mol. Sci. 2018, 19(10), 2913; https://doi.org/10.3390/ijms19102 913

It might be good to highlight the limitation of the current model and how more complex in-vitro systems could be generated in the future. I do not think that it would diminish the value of the present study in any way nor would I expect the general nature of the data to change in a more complex system.  

Nanoparticles themselves have the ability to cross the endothelial barrier

Scientific Reports 7, Article number: 707 (2017) (Ho, Adriani, et al)

It would be interesting to learn whether the authors observed trans-barrier (migration/transport) across both cell layers (endothelial/epithelial) or maybe the nanoparticles accumulated in between the two cell layer. 

However, no additional experiments need to be performed, if the authors happen to have data it would certainly not harm to reference inside the manuscript to this observations.“

-       The references suggested by the reviewer have been added to the discussion on page 12 of the manuscript.

-       I agree with the reviewer´s interest whether we observed a trans-barrier translocation of nanoparticles across the two cell layers. These kind of studies have already been conducted for the alveolar-capillary barrier model. For proper translocation studies a different experimental setup would have been needed since the pore-size of the transwell filters in this study would most likely hamper/decelerate translocation of the nanoparticles through the membrane. A more detailed explanation is added to the discussion at line 312 with respective citations.

Reviewer 2 Report

The authors describe a barrier model by combining the enterocyte cell line (Caco-2) and microvascular endothelial cell line (ISO-HAS-1), each of the opposing sides to a transwell filter-membrane. The authors investigated the release of common biomarkers for IBD or inflammatory mediators, IL-8, soluble ICAM-1 and soluble E-selectin. The authors also introduced silica nanoparticles with different surface functionalities, which demonstrated reduction in endothelial microparticles.

Overall, the work done was very interesting and the barrier model could aid in further research in this field, in particular in the IBD area. With that said, there are some comments that should be addressed.

There are many errors within the text and it is suggested that the authors carefully review the document.

In the introduction, the authors do not discuss other model barrier systems. What is currently used to investigate the intestinal blood barrier? If there is no other model, make this more clear. The introduction should clearly state the impact of this work.

Why was silica nanoparticles chosen for this work? This is not clear and puzzling to the reader. With many other nanoparticles, why was this one chosen for this work and what is the significance? TEM could be used to investigate the uptake of the nanoparticles. Where controls done? How do the authors know that nanoparticles were stable in solution? From the methods, the silca nanoparticles were fluorescently labelle with Rhodamin B, why was there no fluorescent imaging done?

The authors mention that exosomes were isolated but I refer the authors to MISEV-2018 guidelines. This document recommends minimal criteria to define exosomes by detection of exosomes markers e.g. CD63, CD9, ALIX, CD86 etc. (https://www.tandfonline.com/doi/full/10.1080/20013078.2018.1535750)

Author Response

Dear Prof. Dr. Jellinger, Dear Ms. Wu, Dear reviewer,

 thank you for your quick response regarding our manuscript submitted to your special issue entitled „Endothelial Dysfunction“ of the „International Journal of Molecular Sciences“ with the title: „An in vitro entero-capillary barrier exhibits an altered inflammatory and exosomal communication pattern after exposure to silica nanoparticles“. The coautors are Jennifer Y. Kasper, M. Iris Hermanns, Annette Kraegeloh, W. Roth, C. James Kirkpatrick, Ronald E. Unger.

Attached you will find our responses to the reviewer´s comments.   

„The authors describe a barrier model by combining the enterocyte cell line (Caco-2) and microvascular endothelial cell line (ISO-HAS-1), each of the opposing sides to a transwell filter-membrane. The authors investigated the release of common biomarkers for IBD or inflammatory mediators, IL-8, soluble ICAM-1 and soluble E-selectin. The authors also introduced silica nanoparticles with different surface functionalities, which demonstrated reduction in endothelial microparticles.

 Overall, the work done was very interesting and the barrier model could aid in further research in this field, in particular in the IBD area. With that said, there are some comments that should be addressed.“

Comment 1:

„There are many errors within the text and it is suggested that the authors carefully review the document.“

-       The manuscript has been carefully read and reviewed by english native speaking individuals to eliminate spelling and grammatical as well typing errors.

  Comment 2:

 â€žIn the introduction, the authors do not discuss other model barrier systems. What is currently used to investigate the intestinal blood barrier? If there is no other model, make this more clear. The introduction should clearly state the impact of this work.“

 -       The introduction has been modified to address the reviewer´s request as follows (line 56):

A number of in vivo models to study iMV behavior upon systemic inflammation have been described using Wistar rats [13]. In vitro studies examining ICAM/E-Selectin induction in monocultures of primary isolated intestinal microvascular endothelial cells (IMEC) have also been described [14]. However, to date, it appears that only a single in vitro coculture model mimicking the intestinal barrier consisting of microvascular endothelial cells in combination with enterocytes has been described [15]. No in vitro model for IBD has been developed. Thus, a model mimicking the in vivo intestinal entero-capillary barrier would be highly useful in understanding exacerbated inflammatory progressions that result in barrier disruption in IBD patients. Furthermore, in vitro models mimicking disease situations can be indispensable in screening and evaluating the efficacy of novel therapeutic strategies e.g. a nanoparticle-mediated drug delivery specifically targeting the endothelial lining [16]. 

Comment 3:

„Why was silica nanoparticles chosen for this work? This is not clear and puzzling to the reader. With many other nanoparticles, why was this one chosen for this work and what is the significance? TEM could be used to investigate the uptake of the nanoparticles. Where controls done? How do the authors know that nanoparticles were stable in solution? From the methods, the silca nanoparticles were fluorescently labelle with Rhodamin B, why was there no fluorescent imaging done?“

-       Silica nanoparticles have been chosen for several reasons. On the one hand aSNPs are considered as potential drug delivery vehicles and on the other hand, these NPs have been used extensively and characterized by the authors in previous studies concerning the interactions of these particles with the in vitro alveolar-capillary barrier model. We agree that TEM would be a good option for high resolution imaging to detect the aSNPs within the cells. However, silica is very difficult to detect by TEM and optimal conditions could not be established. Fluorescence imaging and localization of the aSNPs within the two types of cells used in this study has been previously reported. The respective explanation and references are added in the manuscript as follows (line 315): Additionally, it is known that nanoparticles themselves have the ability to cross the endothelial barrier. Recent studies by Ho and coworkers described methods to investigate the extravasation of nanoparticles [50]. aSNP uptake and translocation experiments have been described for ISO-HAS-1 using the lung equivalent in vitro model of the alveolar-capillary barrier [48]. In the transwell model, aSNPs were able to cross the semi-permeable transwell filtermembrane with a pore size of 400 nm. However, low pore size may decelerate aSNP translocation. Furthermore, cellular uptake experiments indicated a colocalization of internalized aSNPs with flotillin bearing endosomes [47, 48]. Thus, additional studies are necessary to obtain a better understanding of the intracellular uptake and translocation of aSNPs in the gut blood barrier in vitro.

-       The respective controls

The authors mention that exosomes were isolated but I refer the authors to MISEV-2018 guidelines. This document recommends minimal criteria to define exosomes by detection of exosomes markers e.g. CD63, CD9, ALIX, CD86 etc. (https://www.tandfonline.com/doi/full/10.1080/20013078.2018.1535750)

-       We agree with the reviewer`s comments regarding the characterization of exosomes.  The results described in the present study were carried out in 2015 at which time the MISEV-2018 guidlines were not yet published. We have added a sentence in the manuscript to address this. The studies presented in this manuscript indicate that exosomes may be responsible for the results observed, however, we also state that further and more detailed and comprehensive studies are necessary to determine what type of communication takes place between the different cell types.  We have added this to the discussion section in line 294 and have made reference to the MISEV-2018 guidelines as follows: No distinction was made in the present study between smaller exosomes or larger microvesicles. These need to be evaluated in future studies to determine exosomal size and surface markers as well as their cargo. Furthermore, additional studies should focus on a more detailed characterization that includes imaging via transmission electron microscopy and the evaluation of exosomal markers e.g. CD63, CD9, ALIX or CD86, which are recommended as minimal criteria to define exosomes according to the MISEV-2018 guidelines [46].

Round  2

Reviewer 2 Report

The authors describe a barrier model by combining the enterocyte cell line (Caco-2) and microvascular endothelial cell line (ISO-HAS-1), each of the opposing sides to a transwell filter-membrane. The authors investigated the release of common biomarkers for IBD or inflammatory mediators, IL-8, soluble ICAM-1, and soluble E-selectin. Silica nanoparticles with different surface functionalities were introduced to the model and demonstrated a reduction in endothelial microparticles. Overall, the work done was very interesting and the barrier model could aid in further research in this field, in particular in the IBD area. Since the first review round, authors have addressed previous comments and it is believed that the manuscript has been significantly improved.